The Modern Research Data Portal: a design pattern for networked, data-intensive science

Chard Kyle 1 2
Dart Eli 3
Foster Ian foster@anl.gov 1 2
Shifflett David 1 2
Tuecke Steven 1 2
Williams Jason 1 2
1 University of Chicago , Chicago , IL , United States of America
2 Argonne National Laboratory , Lemont , IL , United States of America
3 Energy Sciences Network, Lawrence Berkeley National Laboratory , Berkeley , CA , United States of America
Ludäscher Bertram
Electronic publication date: 2018 Jan 15
Publication date: 2018
Volume: 4
Electronic Location ID: e144
Received 2017 Aug 26; Accepted 2017 Dec 19
Copyright: ©2018 Chard et al.
Copyright year: 2018
Copyright holder: Chard et al.
License: This is an open access article distributed under the terms of the Creative Commons Attribution License, which permits unrestricted use, distribution, reproduction and adaptation in any medium and for any purpose provided that it is properly attributed. For attribution, the original author(s), title, publication source (PeerJ Computer Science) and either DOI or URL of the article must be cited.
License URL: https://creativecommons.org/licenses/by/4.0/

Keywords: Portal, High-speed network, Globus, Science DMZ, Data transfer node

Funding: United States National Science Foundation ACI-1148484 Department of Energy’s Office of Advanced Scientific Computing Research DE-AC02-06CH11357 This work was supported by the United States National Science Foundation (ACI-1148484) and Department of Energy’s Office of Advanced Scientific Computing Research (DE-AC02-06CH11357). The funders had no role in study design, data collection and analysis, decision to publish, or preparation of the manuscript.

==============================
We describe best practices for providing convenient, high-speed, secure access to large data via research data portals. We capture these best practices in a new design pattern, the Modern Research Data Portal, that disaggregates the traditional monolithic web-based data portal to achieve orders-of-magnitude increases in data transfer performance, support new deployment architectures that decouple control logic from data storage, and reduce development and operations costs. We introduce the design pattern; explain how it leverages high-performance data enclaves and cloud-based data management services; review representative examples at research laboratories and universities, including both experimental facilities and supercomputer sites; describe how to leverage Python APIs for authentication, authorization, data transfer, and data sharing; and use coding examples to demonstrate how these APIs can be used to implement a range of research data portal capabilities. Sample code at a companion web site, https://docs.globus.org/mrdp, provides application skeletons that readers can adapt to realize their own research data portals.

Introduction

The need for scientists to exchange data has led to an explosion over recent decades in the number and variety of research data portals: systems that provide remote access to data repositories for such purposes as discovery and distribution of reference data, the upload of new data for analysis and/or integration, and data sharing for collaborative analysis. Most such systems implement variants of a design pattern (Gamma et al., 1994) that we term the legacy research data portal (LRDP), in which a web server reads and writes a directly connected data repository in response to client requests.

The relative simplicity of this structure has allowed it to persist largely unchanged from the first days of the web. However, its monolithic architecture—in particular, its tight integration of control channel processing (request processing, user authentication) and data channel processing (routing of data to/from remote sources and data repositories)—has increasingly become an obstacle to performance, usability, and security, for reasons discussed below.

An alternative architecture re-imagines the data portal in a much more scalable and performant form. In what we term here the modern research data portal (MRDP) design pattern, portal functionality is decomposed along two distinct but complementary dimensions. First, control channel communications and data channel communications are separated, with the former handled by a web server computer deployed (most often) in the institution’s enterprise network and the latter by specialized data servers connected directly to high-speed networks and storage systems. Second, responsibility for managing data transfers, data access, and sometimes also authentication is outsourced to external, often cloud-hosted, services. The design pattern thus defines distinct roles for the web server, which manages who is allowed to do what; data servers, where authorized operations are performed on data; and external services, which orchestrate data access.

In this article, we first define the problems that research data portals address, introduce the legacy approach, and examine its limitations. We then introduce the MRDP design pattern and describe its realization via the integration of two elements: Science DMZs (Dart et al., 2013) (high-performance network enclaves that connect large-scale data servers directly to high-speed networks) and cloud-based data management and authentication services such as those provided by Globus (Chard, Tuecke & Foster, 2014). We then outline a reference implementation of the MRDP design pattern, also provided in its entirety on the companion web site, https://docs.globus.org/mrdp, that the reader can study—and, if they so desire, deploy and adapt to build their own high-performance research data portal. We also review various deployments to show how the MRDP approach has been applied in practice: examples like the National Center for Atmospheric Research’s Research Data Archive, which provides for high-speed data delivery to thousands of geoscientists; the Sanger Imputation Service, which provides for online analysis of user-provided genomic data; the Globus data publication service, which provides for interactive data publication and discovery; and the DMagic data sharing system for data distribution from light sources. We conclude with a discussion of related technologies and summary.

The research data portal

The exchange of data among researchers is a fundamental activity in science (Borgman, 2012; Hey, Tansley & Tolle, 2009; Tenopir et al., 2011), and the use of computer networks for that purpose dates back to the earliest days of the Internet. Indeed, it was the need for convenient data exchange that drove Tim Berners-Lee to invent the web in 1989 (Berners-Lee, 1989). He defined an architecture of extreme simplicity. A computer with access to both a data repository and the Internet runs a web server application that performs upload and download operations on its local data repository based on client requests. Users issue such requests using a client program—typically a web browser, which provides a graphical user interface (GUI). A uniform naming scheme for data objects makes it easy to share names, for example by embedding them in specially formatted documents (web pages).

Web technologies have since evolved tremendously. For example, web servers can now support client authentication and authorization, link to databases for efficient navigation of large repositories, and support Common Gateway Interface (CGI) access to server-side computation. Web browsers run powerful Javascript and support asynchronous requests to remote servers.

Yet as shown in Fig. 1, most systems used to exchange research data today are not so different from that first web server. In particular, a single server handles request processing, data access, authentication, and other functions. It is the simple and monolithic architecture that characterizes this legacy research data portal (LRDP) design pattern that has allowed its widespread application and its adaptation to many purposes. Under names such as portals (Russell et al., 2001), science gateways (Wilkins-Diehr et al., 2008; Lawrence et al., 2015), and hubs (Klimeck et al., 2008; McLennan & Kennell, 2010), LRDP instances variously support access to small and large scientific data collections, data publication to community repositories, online data analysis and simulation, and other scientific workflows concerned with managing the flow of scientific data between remote clients and a central server.

Figure 1 The legacy research data portal (LRDP) architecture.

The portal web server runs all portal services and also manages all data objects and their transport to and from the wide area network (WAN).

A confluence of three factors has now rendered the monolithic LRDP architecture increasingly problematic for its intended purpose, as we now discuss.

The first concerns performance. Both data volumes and network speeds have grown tremendously over the past decade, at rates far faster than can be supported by monolithic web services applications running on individual servers connected to enterprise networks and secured by enterprise firewalls. In response, network architectures are proliferating to allow high-speed access to research data over separate data channels that are specifically engineered for high-performance data services. However, these developments do not mesh well with the LRDP architecture. The LRDP server typically cannot be moved outside the institutional firewall because it is a complex web services application with sensitive information (e.g., user databases) that must be defended using correspondingly complex security protections. Also, the LRDP model forces all data objects through the web server software stack, limiting performance and scalability.

The second is an increasing demand for high data transfer reliability. When download or upload requests involve just a few small files, transient errors are rare and can be handled by having users resubmit requests. When requests involve thousands or more files and gigabytes or terabytes, errors are more problematic. Increasingly, researchers expect the high reliability offered by transfer services such as Globus (Chard, Tuecke & Foster, 2014), which use specialized techniques to minimize the impact of errors. But retrofitting such mechanisms into the LRDP model is challenging.

The third challenge is operational complexity, which in turn can negatively impact development costs, reliability, capabilities offered, and security. Conventional LRDP implementations are developed, deployed, and operated in silos, with each establishing and operating its own implementations of user management, authentication, authorization, and data transfer. This siloed approach is sub-optimal and inefficient, and often results in limited functionality relative to state-of-the-art. For example, few legacy portals enable authentication and authorization with standard web protocols such as OpenID Connect (Sakimura et al., 2014) and OAuth 2 (Hardt, 2012), instead preferring to manage local user and authorization databases. The siloed approach makes it difficult to ensure that best practices, especially with respect to security, are followed. It also increases the burden on administrators to ensure that not only is the portal available but also that updates and security patches are applied quickly.

In summary, while the LRDP model has served the scientific community well for many years, it suffers from fundamental limitations, with the result that portals based on the LRDP model serve the scientific community less well than they otherwise could.

The MRDP design pattern

This discussion of LRDP limitations brings us to the core of this article, namely, the modern research data portal (MRDP) design pattern. This new approach to building research data portals became feasible in around 2015, following the maturation of two independent efforts aimed at decoupling research data access from enterprise networks, on the one hand, and research data management logic from storage system access, on the other. These developments inspired a new approach to the construction of research data portals based on a decomposition of the previously monolithic LRDP architecture into three distinct components:

1. The portal server (a web server like any other) which handles data search and access, mapping between users and datasets, and other web services tasks;

2. A high-performance network enclave that connects large-scale data servers directly to high-performance networks (we use the Science DMZ as an example here); and

3. A reliable, high-performance external data management service with authentication and other primitives based on standard web APIs (we use Globus as an example here).

In the remainder of this section, we describe the role of the Science DMZ and the data servers that reside within it, the role of Globus as a provider of outsourced data management and security services, and the integration of these components to form the MRDP design pattern.

Science DMZ and DTNs

A growing number of research institutions are connected to high-speed wide area networks at high speeds: 10 gigabits per second (Gb/s) or faster. Increasingly, these wide area networks are themselves connected to cloud providers at comparable speeds. Thus, in principle, it should be possible to move data between any element of the national research infrastructure—between institutions, laboratories, instruments, data centers, supercomputer centers, and clouds—with great rapidity.

In practice, real transfers often achieve nothing like these peak speeds. Common reasons for poor performance are the complexity and architectural limitations of institutional networks, as well as complex and inefficient configurations on monolithic server computers. Commodity network devices, firewalls, and other limitations cause performance bottlenecks in the network between the data service and the outside world where clients are located.

Two constructs, the Science DMZ and the Data Transfer Node, are now widely deployed to overcome this problem. As shown in Fig. 2, the Science DMZ overcomes the challenges associated with multi-purpose enterprise network architectures by placing resources that need high-performance connectivity in a special subnetwork that is close (from a network architecture perspective) to the border router that connects the institution to the high-speed wide area network. (The term DMZ, short for demilitarized zone, is commonly used in computer networking to indicate an intermediate role between external and internal networks.) Traffic between those resources and the outside world then has a clean path to the analogous high-performance resources at collaborating institutions.

Figure 2 The MRDP design pattern from a network architecture perspective.

The Science DMZ includes multiple DTNs that provide for high-speed transfer between network and storage. Portal functions run on a portal server, located on the institution’s enterprise network. The DTNs need only speak the API of the data management service (Globus in this case).

A Data Transfer Node (DTN) is a specialized device dedicated to data transfer functions. These devices are typically PC-based Linux servers constructed with high quality components, configured for both high-speed wide area data transfer and high-speed access to local storage resources, and running high-performance data transfer tools such as Globus Connect data transfer software. General-purpose computing and business productivity applications, such as email clients and document editors, are not installed; this restriction produces more consistent data transfer behavior and makes security policies easier to enforce.

The Science DMZ design pattern also includes other elements, such as integrated monitoring devices for performance debugging, specialized security configurations, and variants used to integrate supercomputers and other resources. For example, Fig. 2 shows perfSONAR (Hanemann et al., 2005) performance monitoring devices. But this brief description provides the essential information required for our discussion here. The US Department of Energy’s Energy Sciences Network has produced detailed configuration and tuning guides for Science DMZs and DTNs (ESnet, 2017).

Globus services

Globus provides data and identity management capabilities designed for the research community. These capabilities are delivered via a cloud-hosted software- and platform-as-a-service model, enabling users to access them through their web browser and developers to invoke them via powerful APIs. We describe here Globus capabilities that meet MRDP needs for managing and transferring data (Chard, Tuecke & Foster, 2014) and for authenticating users and authorizing access (Tuecke et al., 2015).

Globus allows data to be remotely managed across its pool of more than 10,000 accessible storage systems (called “endpoints”). A storage system is made accessible to Globus, and thus capable of high performance and reliable data transfer, by installing Globus Connect software. Globus Connect is offered in two versions: Globus Connect Personal for single-user deployments (e.g., a laptop or PC) and Globus Connect Server for multi-user deployments (e.g., a shared server or DTN). Globus Connect Server can be deployed on multiple DTNs associated with a storage system; Globus then uses the pool of DTNs to increase transfer performance, with dynamic failover for increased reliability.

Globus Transfer capabilities provide high performance and reliable third party data transfer. The Globus service manages the entire transfer process, including coordinating authentication at source and destination; establishing a high performance data channel using the GridFTP (Allcock et al., 2005) protocol, with configuration optimized for transfer; ensuring data integrity by comparing source/destination checksums; and recovering from any errors during the transfer. Globus also provides secure (authorized) HTTPS access to (upload/download) data via a web browser or an HTTP command line client (e.g., for small files, inline viewers, or transitional support of LRDP usage models). Globus, by default, enforces the data access permissions represented by the underlying system; however, it also allows these access decisions to be managed through the cloud service. In the latter mode, called Globus Sharing (Chard, Tuecke & Foster, 2014), users may associate user- or group-based access control lists (ACLs) with particular file paths. Globus checks and enforces these ACLs when other users attempt to read or write to those paths.

Globus Auth provides identity and access management platform capabilities. It brokers authentication and authorization interactions between end-users, identity providers, resource servers (services), and clients (e.g., web, mobile, desktop, and command line applications, and other services). It implements standard web protocols, such as OAuth 2 and OpenID Connect, that allow it to be integrated easily with external applications using standard client libraries. These protocols enable third-party applications to authenticate users (using their preferred identity) directly with the chosen identity provider. Globus Auth then returns access tokens that the third-party application can use to validate the user’s identity and to perform actions on behalf of that user, within an agreed upon scope. Globus Auth implements an identity federation model via which diverse identities can be linked, and such that presentation of one identity may support authorization for the set of identities. Integration with Globus Groups (Chard et al., 2016) supports group-based authorization using user-managed groups.

REST APIs allow Globus capabilities to be used as a platform. These APIs are used by client libraries, such as the Globus Python SDK, to support integration in external applications. We leverage the Globus Python SDK in the MRDP reference implementation that we describe in this paper, and in the code examples presented in subsequent sections.

The design pattern in practice

The MRDP design pattern improves on the LRDP architecture in three important ways. While these improvements involve relatively minor changes to the web server logic, they have big implications for how data accesses and transfers are performed.

First, data references. An important interaction pattern in a research data portal is the redirect, in which a client request for information leads to the return of one or more URLs for data files. In the LRDP pattern, these are web URLs that reference files served by the same web server that runs the portal GUI. In the MRDP pattern, the portal instead returns references to data objects served by DTNs located in a Science DMZ separate from the portal web server, as shown in Fig. 2. This approach allows the data objects to be transferred using infrastructure that matches their scale. Also, because the DTN cluster can be expanded (e.g., by adding more or faster DTNs) without changing the interface that it provides to the portal, it can easily be scaled up as datasets and traffic volumes grow, without modifying the portal code.

Second, data access. In the LRDP pattern, the references to data objects are returned to the user in a web page. The user must then access the data objects by clicking on links or by using a web command line client like wget. Neither approach is convenient or reliable when dealing with many files. In the MRDP pattern, references are encapsulated in a data transfer job which can be managed by a cloud-based data transfer service such as Globus. Thus the user can hand off the complexity of managing the correct transfer of perhaps many thousands of files to a data management service that is designed to handle such tasks reliably.

Third, user and group management, and authentication and authorization. In the LRDP pattern, these functions are typically all hosted on the web server. Portal developers must therefore implement workflows for authenticating users, requesting access to data, assembling datasets for download, dynamically authorizing access to data, and checking data integrity (e.g., by providing checksums). In the MRDP pattern, these functions can be outsourced via standard interfaces to external services that provide best-practices implementations.

An important feature of the MRDP design pattern is the use of web service (REST) APIs when accessing external services that provide data management and other capabilities. This approach makes it straightforward to retrofit advanced MRDP features into an existing data portal.

Variants of the basic pattern

In the following, we present a reference implementation of the MRDP design pattern that enables download of data selected on a web page. Many variants of this basic MRDP design pattern have been constructed. For example, the data that users access may come from an experimental facility rather than a data archive, in which case they may be deleted after download. Access may be granted to groups of users rather than individuals. Data may be publicly available; alternatively, access may require approval from portal administrators. A portal may allow its users to upload datasets for analysis and then retrieve analysis results. A data publication portal may accept data submissions from users, and load data that pass quality control procedures or curator approval into a public archive. We give examples of several such variants below, and show that each can naturally be expressed in terms of the MRDP design pattern.

Similarly, while we have described the research data portal in the context of a Science DMZ, in which (as shown in Fig. 2) the portal server and data store both sit within a research institution, other distributions are also possible and can have advantages. For example, the portal can be deployed on the public cloud for high availability, while data sit within a research institution’s Science DMZ to enable direct access from high-speed research networks and/or to avoid cloud storage charges. Alternatively, the portal can be run within the research institution and data served from cloud storage. Or both components can be run on cloud resources.

A reference MRDP implementation

We have developed a reference implementation of the MRDP design pattern, for which open source code is available on the companion web site. This code includes:

• A complete, working portal server, implemented with the Python Flask framework and comprising a web service and web interface, and that uses Globus APIs to outsource data transfer, authentication, and authorization functions.

• Integration with Globus Auth for authentication and authorization.

• Integration with Globus Transfer for browsing and downloading datasets.

• Use of a decoupled Globus endpoint for serving data securely via HTTP or GridFTP.

• An independent analysis service, accessed via a REST API, to demonstrate how a data portal can outsource specific functionality securely.

We review some highlights of this reference implementation here; more details are available on the companion web site.

Figure 3 shows the essential elements of this reference implementation. In addition to the Science DMZ and portal server already seen in Fig. 2, we see the user’s desktop computer or other device on which they run the web browser (or other client) used to access the data portal, plus other elements (identity providers, Globus cloud, other services) that we introduce in the following. Note the “Globus endpoints” (Globus Connect servers) located in the Science DMZ, on other remote storage systems, and (optionally) on the user’s desktop. The cloud-hosted Globus transfer service orchestrates high-speed, reliable GridFTP transfers among these endpoints, under the direction of the portal server.

Figure 3 MRDP basics. Clients (left) authenticate with any one of many identity providers (top) and connect to the portal web server (center) that implements the domain-specific portal logic.

The web server sits behind the institutional firewall (red). The portal responds to client requests by using REST APIs to direct Globus cloud services (top) to operate on research data in the Science DMZ (bottom center) and/or to interact with other services (center right). Data flows reliably and securely between Globus endpoints without traversing the firewall.

As already noted, the portal server is at the heart of the MRDP implementation. It sits behind the institutional firewall, from where it serves up web pages to users, responds to HTTP requests, and issues REST communications to Globus services and optionally other services to implement MRDP behaviors. The latter REST communications are central to the power of the MRDP design pattern, as it is they that let the web server outsource many of the complex tasks associated with portal operations. Only the web server component needs to be provided for a specific MRDP implementation.

Overview of key points

The MRDP design pattern employs a collection of modern approaches to delivering critical capabilities. We review some important points here, with comments on how they can be realized with Globus mechanisms. More specifics are on the companion web site.

Outsource responsibility for determining user identities. Operating an identity provider should not need to be a core competency for a portal administrator. Globus Auth support for the OAuth 2 and OpenID protocols allows for the validation of credentials from any one of a number of identity providers deemed acceptable by the portal: for example, institutional credentials via InCommon (Barnett et al., 2011), ORCID, or Google identities.

Outsource control over who can access different data and services within the portal. Nor should the portal developer or administrator need to be concerned with implementing access control mechanisms. The Globus transfer service can be used to control who is allowed to access data.

Outsource responsibility for managing data uploads and downloads between a variety of locations and storage systems. Reliable, efficient, and secure data transfer between endpoints is a challenging task to do well. Again, simple APIs allow this task to be handed off to the Globus transfer service. (A portal can also leverage Globus HTTPS support to provide web-based download and inline viewers.) Thus, the portal does not need to provide its own (typically unreliable and/or slow) data download client, as some portals do.

Leverage standard web user interfaces for common user actions. The implementation of the portal’s web interface can be simplified via the use of standard components. Globus provides web helper pages for such tasks as selecting endpoints, folders, and files for transfers; managing group membership; and logging out.

Dispatch tasks to other services on behalf of requesting users. Good programming practice often involves decomposing a complex portal into multiple services. Globus Auth dependent grant flows (an OAuth concept) enable a portal to allow other services to operate on managed data. We discuss below how the reference implementation uses this mechanism to allow an external data analysis service to access a user’s data.

Log all actions performed by users for purposes of audit, accounting, and reporting. The portal should store a historical log of all actions performed by the portal and its users such that others can determine what data has been accessed, when, and by whom. Again, these functions can be outsourced to a cloud service.

Diving into code

The reference implementation code is for a portal that allows users to browse for data of interest and then request those data for download. The portal makes the data available via four simple steps: (1) create a shared endpoint; (2) copy the requested data to that shared endpoint; (3) set permissions on the shared endpoint to enable access by the requesting user, and email the user a URL that they can use to retrieve data from the shared endpoint; and ultimately (perhaps after days or weeks), (4) delete the shared endpoint.

Listing 1 presents a function rdp that uses the Globus Python SDK to implement these actions. This code is somewhat simplified: it does not perform error checking and does not integrate with the web server that our data portal also requires, but otherwise it is a complete implementation of the core portal logic–in just 42 lines of code. In the following, we first review some Globus background and then describe some elements of this program.

Endpoints

Figure 3 shows several Globus endpoints. Each endpoint implements the GridFTP protocol (thick green arrows) for high-speed endpoint-to-endpoint data transfer and the HTTPS protocol (blue arrows) to enable access to endpoint storage from a web client. Endpoints are managed by the Globus Transfer service via the GridFTP control channel: the thin green dashed lines.

In order for Globus Transfer to perform operations on an endpoint filesystem, it must have a credential to authenticate to the endpoint as a specific local user. The process of providing such a credential to the service is called endpoint activation. Endpoints typically require credentials from one or more specific identity providers. A Globus Connect Server can be configured with a co-located MyProxy OAuth (Basney & Gaynor, 2011) server to allow local user accounts to be used for authentication. Alternatively, endpoints may be configured to use one of the supported Globus Auth identity providers. In this case, endpoints contain a mapping between identity provider identities and local user accounts. Each Globus endpoint (and each user) is named by a universally unique identifier (UUID).

Identities and credentials

We see from Fig. 3 that the MRDP design pattern involves interactions among numerous entities: user, web client, Globus services, portal server, Globus endpoints, and perhaps other services as well. Each interaction requires authentication to determine the originator of the request and access management to determine whether the request should be granted.

Access management decisions are complicated by the fact that a single user may possess, and need to use, multiple identities from different identity providers. For example, a user Jane Doe with an identity jane@uni.edu at her university and another identity jdoe@lab.gov at a national laboratory may need to use both identities when transferring a file from endpoint A at uni.edu to endpoint B at lab.gov.

The Globus Auth identity and access management service manages these complexities. When Jane uses a web client to log in to Globus, Globus Auth requests that she verify her identity via one of her linked identity providers (e.g., by obtaining and checking a password). This verification is performed via an OAuth 2 redirection workflow to ensure that the password used for verification is never seen by Globus Auth. Upon successful authentication, Globus Auth creates a set of unique, fixed-term access tokens and returns them to the requesting client. Each access token specifies the time period and purpose for which it can be used (e.g., transferring data, accessing profile information). The web client can then associate a token with subsequent REST requests—for example a transfer request to Globus Transfer—to demonstrate that Jane has verified her identity.

Let’s say that Jane now uses the Globus Transfer web client to request a file transfer from endpoint A at uni.edu to endpoint B at lab.gov. Endpoint A requires a uni.edu access token, which the web client can provide due to the initial authentication. Endpoint B requires a lab.gov access token, which the web client does not possess. The Globus Transfer web client will then ask Globus Auth to invoke a further OAuth 2 operation to verify that identity and obtain a second access token.

The MRDP implementation employs Globus Sharing to control data access. To use this feature, a user authorized to operate on the existing endpoint first creates a shared endpoint, designating the existing endpoint and folder, and then grants read and/or write permissions on the shared endpoint to the Globus user(s) and/or group(s) that they want to access it. Those users can then access that endpoint like any other endpoint, but do not need a local account.

This construct is illustrated in Fig. 4. Bob has enabled sharing of folder ˜/shared_dir on Regular endpoint to create Shared endpoint, and then granted Jane access to that shared endpoint. Jane can then use Globus to read and/or write files in the shared folder, depending on what rights she has been granted.

Figure 4 A shared endpoint scenario.

The rdp function

The three arguments to the function rdp in Listing 1 can be understood in terms of Fig. 4. Those arguments are, in turn, the UUID for the endpoint on which the shared endpoint is to be created, the name of the folder on that endpoint from which the contents of the shared folder are to be copied, and the email address for the user who is to be granted access to the shared endpoint. In this case we use the Globus endpoint named “Globus Tutorial Endpoint 1” and sample data available on that endpoint.

 rdp('ddb59aef-6d04-11e5-ba46-22000b92c6ec',       '~/share/godata/',       'jane@uni.edu')

The code in Listing 1 uses the Globus Python SDK to create, manage access to, and delete the shared endpoint, as follows. It first creates a TransferClient and an AuthClient object—classes provided by the Globus Python SDK for accessing the Globus Transfer and Globus Auth services, respectively. Each class provides a rich set of methods for accessing the various resources defined in the REST APIs. We then use the SDK function endpoint_autoactivate to ensure that the portal has a credential that permits access to the endpoint identified by host_id.

In Step 1(a), we use the Globus SDK function operation_mkdir to create a directory (named, in our example call, by a UUID) on the endpoint with identifier host_id. Then, in Step 1(b), the SDK function create_shared_endpoint is used to create a shared endpoint for the new directory. At this point, the new shared endpoint exists and is associated with the new directory, but only the creating user has access.

In Step 3, we first use the Globus SDK function get_identities to retrieve the user identifier associated with the supplied email address; this is the user for whom sharing is to be enabled. (If this user is not known to Globus, an identity is created.) We then use the function add_endpoint_acl_rule to add to the new shared endpoint an access control rule that grants the specified user read-only access to the endpoint.

As our add_endpoint_acl_rule request specifies an email address, an invitation email is sent. At this point, the user is authorized to download data from the new shared endpoint. The shared endpoint is typically left operational for some period and then deleted, as shown in Step 4. Note that deleting a shared endpoint does not delete the data that it contains. The portal admin may want to retain the data for other purposes. If not, we can use the Globus SDK function submit_delete to delete the folder.

         Listing 1: Globus code to implement MRDP design pattern _________________________________________________________________ from globus_sdk import TransferClient, TransferData from globus_sdk import AuthClient import sys, random, uuid def rdp(host_id,     # Endpoint for shared endpoint         source_path, # Directory to copy data from         email):      # Email address to share with     tc = TransferClient()     ac = AuthClient()     tc.endpoint_autoactivate(host_id)     # (1) Create shared endpoint:     # (a) Create directory to be shared     share_path = '/~/' + str(uuid.uuid4()) + '/'     tc.operation_mkdir(host_id, path=share_path)     # (b) Create shared endpoint on directory     shared_ep_data = {       'DATA_TYPE': 'shared_endpoint',       'host_endpoint': host_id,       'host_path': share_path,       'display_name': 'RDP  shared  endpoint',       'description': 'RDP  shared  endpoint'     }     r = tc.create_shared_endpoint(shared_ep_data)     share_id = r['id']     # (2) Copy data into the shared endpoint     tc.endpoint_autoactivate(share_id)     tdata = TransferData(tc, host_id, share_id,         label='RDP  copy', sync_level='checksum')     tdata.add_item(source_path, '/', recursive=True)     r = tc.submit_transfer(tdata)     tc.task_wait(r['task_id'], timeout=1000,                  polling_interval=10)     # (3) Enable access by user     r = ac.get_identities(usernames=email)     user_id = r['identities'][0]['id']     rule_data = {       'DATA_TYPE': 'access',       'principal_type': 'identity', # Grantee is       'principal': user_id,         #  a user.       'path': '/',                  # Path is /       'permissions': 'r',           # Read-only       'notify_email': email,        # Email invite       'notify_message':             # Invite msg            'Requested  data  are  available.'     }     tc.add_endpoint_acl_rule(share_id, rule_data)     # (4) Ultimately, delete the shared endpoint     tc.delete_endpoint(share_id) _________________________________________________________________

Data transfer

We skipped over Step 2 of Listing 1 in the preceding discussion. That step requests Globus to transfer the contents of the folder source_path to the new shared endpoint. (The transfer in the code is from the endpoint on which the new shared endpoint has been created, but it could be from any Globus endpoint that the portal administrator is authorized to access.) The code assembles and submits the transfer request, providing the endpoint identifiers, source and destination paths, and (as we want to transfer a directory) the recursive flag. It then waits for either the transfer to complete or for a specified timeout to elapse. In practice, the code should check for the response code from the task_wait call and then repeat the wait or terminate the transfer on error or timeout.

Web and command line interfaces

Having received an email invitation to access the new shared endpoint, a user can click on the embedded URL to access that endpoint via the Globus web interface. The user can then transfer the data from that endpoint to any other endpoint to which they are authorized to transfer.

The Globus web interface also provides access to all other Globus functions, allowing users to create shared endpoints, manage access control to those endpoints, and delete a shared endpoint when it is no longer needed. Users may also use the Python SDK or REST APIs to perform these same actions programmatically. In addition, a Globus command line interface (CLI), implemented via the Python SDK, can be used to perform the operations just described.

Figure 5 Example MRDP showing a list of computed graphs for a user and (inset) one of these graphs.

Completing the MRDP portal server

Our MRDP reference implementation consists of a simple research data portal server that allows users to sign up, log in, select datasets from a set of temperature record datasets, and then either download or request the creation of graphs for the selected datasets. This complete implementation is some 700 lines of code, plus 500 lines for an example service that we discuss in the next section. A screenshot of a deployed version of this code is shown in Fig. 5.

The portal server implementation uses the Python Flask web framework, a system that makes it easy to create simple web servers. The core Flask code defines, via @-prefixed decorators, what actions should be performed when a user accesses specific URLs. These views support both authentication and the core MRDP logic as well as additional functionality for managing profiles. Several views generate template-based web pages that are customized for user interactions.

The OAuth 2 authentication workflow proceeds as follows. When logging in, (login()) the user is redirected to Globus Auth to authenticate with their identity provider. The resulting access code is returned to the portal, where it is exchanged for access tokens that can be used to interact with Globus Auth or other dependent services (e.g., Globus Transfer). Access tokens are then validated and introspected to obtain user information.

The portal provides two methods for accessing data. The first method allows users to download raw datasets from a list of available datasets. In this case, it makes a call to Globus Transfer to retrieve a real-time listing of directories in its associated endpoint. The second method allows users to create graphs dynamically, based on selected datasets (by year and identifier), as described in the following subsection. The resulting graph is stored in the portal’s shared endpoint in a directory accessible only to the requesting user. Irrespective of which method is used, users may download data (i.e., raw datasets or computed graphs) via HTTP (using a direct URL to the object on the portal’s endpoint) or by transferring them to another endpoint. In the later case, the user is prompted for a destination endpoint. Rather than re-implement endpoint browsing web pages already offered by Globus, the portal instead uses Globus “helper pages” to allow users to browse the endpoint. The helper page returns the endpoint ID and path to the portal. The portal then starts the transfer using the Globus Transfer API.

The following code, from the portal implementation file portal/view.py, specifies that when a user visits http://localhost:5000/login, the user should be redirected to the authcallback URL suffix.

@app.route('/login', methods=['GET']) def login():      """Send  the  user  to  Globus  Auth."""      return redirect(url_for('authcallback'))

The URL http://localhost:5000/authcallback in turn calls the authcallback function, shown in Listing 2, which uses the OAuth 2 protocol to obtain access tokens that the portal can subsequently use to interact with Globus Auth or dependent services (e.g., Globus Transfer or the graph service.) The basic idea is as follows. First, the web server redirects the user to authenticate using Globus Auth. The redirect URL includes the URL to return to (http://localhost:5000/authcallback) after the user has authenticated. The response includes a auth code parameter which can be unpacked and then swapped for access tokens by contacting Globus Auth and specifying the scopes needed by the portal. Finally, the resulting access tokens are returned to the portal in a JSON object which also includes information about the user’s identity.

Listing 2:  The authcallback function interacts with Globus Auth to obtain access tokens for the server. ________________________________________________________________________________ @app.route('/authcallback', methods=['GET']) def authcallback():   # Handles the interaction with Globus Auth   # Set up our Globus Auth/OAuth 2 state   redirect_uri = url_for('authcallback',   _external=True)   client = load_portal_client()   client.oauth2_start_flow_authorization_code(redirect_uri,refresh_tokens=True)   # If no "code" parameter, we are starting   # the Globus Auth login flow   if 'code' not in request.args:     auth_uri = client.oauth2_get_authorize_url()     return redirect(auth_uri)   else:     # If we have a "code" param, we're coming     # back from Globus Auth and can exchange     # the auth code for access tokens.     code = request.args.get('code')     tokens = client.oauth2_exchange_code_for_tokens(code)     id_token = tokens.decode_id_token(client)     session.update(       tokens=tokens.by_resource_server,       is_authenticated=True,       name=id_token.get('name', ''),       email=id_token.get('email', ''),       project=id_token.get('project', ''),       primary_username=id_token.get('preferred_username'),       primary_identity=id_token.get('sub'),     )     return redirect(url_for('transfer'))     ___________________________________________________________________________

The last line returns, redirecting the web browser to the portal’s transfer page, as shown in Fig. 6.

Figure 6 A portion of the MRDP reference implementation, showing the five user options at top (each mapped to a “route” in the code) and two of the available datasets.

A request to transfer files requires that the user first select the dataset(s) to be transferred and then specify the destination endpoint and location for the dataset(s). Listing 3 implements these behaviors. First, the code checks that the user has selected datasets on the transfer web page. Then, the code redirects the user to https://www.globus.org/app/browse-endpoint, one of the web helper pages that Globus operates to simplify MRDP implementation. The browse endpoint helper page returns the endpoint ID and path to which the user wants to transfer the selected dataset(s). The submit_transfer function (not shown here) uses the endpoint ID and path to execute a Globus transfer request using code similar to Step 2 in Listing 1.

Listing 3: The transfer() function from the web server reference implementa- tion. ________________________________________________________________________________ @app.route('/transfer', methods=['GET', 'POST']) @authenticated def transfer():   if request.method  == 'GET':     return render_template('transfer.jinja2', datasets=datasets)   if request.method  == 'POST':     # Check that file(s) have been selected for transfer     if not request.form.get('dataset'):       flash('Please  select  at  least  one  dataset.')       return redirect(url_for('transfer'))     params = {       'method': 'POST',       'action': url_for('submit_transfer', _external=True, _scheme='https'),       'filelimit': 0,       'folderlimit': 1     }     browse_endpoint =        'https://www.globus.org/app/browse-endpoint?{}'.format(urlencode(params))     # Save submitted form to session     session['form'] = {       'datasets': request.form.getlist('dataset')     }     # Send to Globus to select a destination endpoint using     # the Browse Endpoint helper page.     return redirect(browse_endpoint)     ____________________________________________________________________________

Invoking other services

The final element of the MRDP design pattern that we discuss here is the invocation of other services, as shown by the arrow labeled REST from the portal server to Other services in Fig. 3. Such calls might be used in an MRDP instance for several reasons. You might want to organize your portal as a lightweight front end (e.g., pure Javascript) that interacts with one or more remote backend (micro)services. You might want to provide services that perform subsetting, quality control, data cleansing, or other analyses before serving data. Another reason is that you might want to provide a public REST API for the main portal machinery, so that other app and service developers can integrate with and build on your portal.

Our reference implementation illustrates this capability. The data portal skeleton allows a client to request that datasets be graphed (graph()). It does not perform those graphing operations itself but instead sends a request to a separate Graph service. The request provides the names of the datasets to be graphed. The Graph service retrieves these datasets from a specified location, runs the graphing program, and uploads the resulting graphs to a dynamically created shared endpoint for subsequent retrieval. The reference implementation includes a complete implementation of the Graph service, showing how it manages authentication and data transfer with Globus APIs.

Examples of the MRDP design pattern

We briefly present five examples of large-scale implementations of the MRDP design pattern. We present performance results for several of these examples in the next section.

The NCAR Research Data Archive

The Research Data Archive (RDA) operated by the US National Center for Atmospheric Research at http://rda.ucar.edu contains more than 600 data collections, ranging in size from megabytes to tens of terabytes. These collections include meteorological and oceanographic observations, operational and reanalysis model outputs, and remote sensing datasets to support atmospheric and geosciences research, along with ancillary datasets, such as topography/bathymetry, vegetation, and land use. The RDA data portal allows users to browse and search data catalogs, and then download selected datasets to their personal computer or HPC facility.

RDA users are primarily researchers at federal and academic research laboratories. In 2016 alone, more than 24,000 people downloaded more than 1.9 petabytes. The RDA portal thus requires robust, scalable, maintainable, and performant implementations of a range of functions, some domain-independent (e.g., user identities, authentication, and data transfer) and others more domain-specific (e.g., a catalog of environmental data collections).

RDA uses the techniques described previously to implement the MRDP design pattern, except that they do not currently outsource identity management. The use of the MRDP design pattern for RDA allows for vastly increased scalability in terms of dataset size, and much lower human effort for managing the transfer of multi-terabyte datasets to computing centers for analysis.

Sanger imputation service

This service, operated by the Wellcome Trust Sanger Institute at https://imputation.sanger.ac.uk, allows you to upload files containing genome wide association study (GWAS) data from the 23andMe genotyping service and receive back the results of imputation and other analyses that identify genes that you are likely to possess based on those GWAS data (McCarthy et al., 2016). This service uses Globus APIs to implement a variant of the MRDP design pattern, as follows.

A user who wants to use the service first registers an imputation job. As part of this process, they are prompted for their name, email address, and identity, and the type of analysis to be performed. The portal then requests Globus to create a shared endpoint, share that endpoint with the identity provided by the user, and email a link to this endpoint to the user. The user clicks on that link to upload their GWAS data file and the corresponding imputation task is added to the imputation queue at the Sanger Institute. Once the imputation task is completed, the portal requests Globus to create a second shared endpoint to contain the output and to email the user a link to that new endpoint for download. The overall process differs from that of Listing 1 only in that a shared endpoint is used for data upload as well as download.

Petrel, a user-managed data sharing portal

Argonne National Laboratory’s Petrel (http://petrel.alcf.anl.gov/) implements a specialized research data portal that allows users to request a space allocation and then upload, download, organize, and share data within that allocated space.

Petrel uses Globus to implement the MRDP model on top of storage provided by Argonne. It implements a simple workflow (based on Globus Groups) for users to request an allocation. It then, using code similar to Listing 1, creates a directory for the allocation and a shared endpoint to manage the allocation, and assigns the requesting user as the “manager” of that shared endpoint. The user can then manage their allocation as if it were storage on their personal resources, uploading and downloading data, and selecting who may access (read/write) paths in that storage.

Scalable data publication

Globus data publication (Chard et al., 2015) enables researchers to publish and access data in user-managed collections. Following the MRDP design pattern, the system is implemented as a cloud-hosted service with data storage provided by decoupled Globus endpoints. Users define their own publication collections by specifying a Globus endpoint on which data are to be stored. The publication service then manages the workflows for submitting a dataset, associating a persistent identifier with the dataset, and recording user-supplied metadata. Permitted users can then publish data in the collection by following this workflow. Once published, other users are then able to discover and download published data.

The service uses Globus transfer to manage data. Upon submission of a new dataset, the publication service creates a unique directory in the collection’s shared endpoint and shares it (with write access) to the submitting user. After the submission is complete, the service removes the write permission (in effect making the endpoint immutable) and proceeds through the submission and curation workflows, by changing permissions on the directory. When the submission is approved for publication, the service modifies permissions to match the collection’s policies. It now acts as a typical MRDP, providing access to published data for discovery and download.

Data delivery at Advanced Photon Source

The Advanced Photon Source (APS) at Argonne National Laboratory, like many experimental facilities worldwide, serves thousands of researchers every year, many of whom collect data and return to their home institution. In the past, data produced during an experiment was invariably carried back on physical media. However, as data sizes have grown and experiments have become more collaborative, that approach has become less effective. Data transfer via network is preferred; the challenge is to integrate data transfer into the experimental workflow of the facility in a way that is automated, secure, reliable, and scalable.

The DMagic system De Carlo (2017) does just that. DMagic integrates with APS administrative and facility systems to deliver data to experimental users. Before the experiment begins, it creates a shared endpoint on a large storage system maintained by Argonne’s computing facility. DMagic then retrieves the list of approved users for the experiment and adds permissions for those users to the shared endpoint. It then monitors the experiment data directory at the APS experimental facility and copies new files automatically to that shared endpoint, from which they can be retrieved by any approved user.

Evaluation of MRDP adoption

The advantages of using high performance data transfer protocols over traditional protocols (e.g., HTTP, SCP) have been well explored (Mattmann et al., 2006; Rao et al., 2016; Subramoni et al., 2010). The benefits of building upon a professionally managed platform are also well established (Cusumano, 2010). Thus, rather than comparing MRDP performance with that of legacy approaches, we examine adoption of the MRDP pattern in real-world deployments. Specifically, we explore usage of four MRDPs: (1) NCAR’s Research Data Archive; (2) the Sanger imputation service; (3) Petrel data sharing service; and (4) Globus data publication (Publish).

Table 1 presents for each portal the number of transfers, number of unique users, total data transferred, and average transfer rates. Figures 7 and 8 show the total data transferred per day and file size vs. transfer rate for individual transfers, respectively, for three of these MRDPs. In all cases we report only successfully completed transfers. (Ongoing transfers, and transfers canceled by users, or by Globus due to errors, are not included.) These results highlight the large amounts of data that can be handled by such portals. In fact, all but one moves in excess of 1 TB per day. (The peak transfer volume of more than 1 PB recorded for Petrel is an artifact of our plotting routine, which assigns the bytes associated with each transfer to the day on which the transfer completes.) These graphs also highlight the wide range of MRDP characteristics that we encounter “in the wild”: for example, whether source or sink, the scales of data handled, achieved rates, and (indirectly) total transfer times.

Table 1 Usage summary for four MRDP deployments: total operations, unique users, total data, and average transfer rate, each for both outgoing and incoming transfers.

	Operations	Users	Data (TB)	Rate (Mbit/s)	
Name	Out	In	Out	In	Out	In	Out	In	
RDA	5,550	1	327	1	308.7	0.0	15.5	1.1	
Sanger	4,849	4,294	343	361	240.3	48.0	45.1	47.54	
Petrel	34,403	16,807	217	100	654.0	1,030.0	142.4	135.0	
Publish	3,306	2,997	107	83	7.1	7.1	17.0	131.8	

Figure 7 Input/output transfer volumes per day for three MRDP instances.

The numbers on the x axis represent days prior to March 9, 2017.

Figure 8 Transfer rate vs. data size for three MRDP instances.

(A–B) show (A) all transfers for the RDA instance and (B) only transfers from RDA to the National Energy Research Supercomputing Center (NERSC), a high performance computing center in Berkeley, California. (We make the points in the latter figure larger, as there are fewer of them.) (C–D) show all transfers for the (C) Sanger and (D) Petrel instances. Each point in each scatter plot represents a single transfer request, which may involve many files. Incoming and outgoing transfers are distinguished; RDA has only one incoming transfer, which we highlight with a circle.

The remarkable variation in transfer performance seen in Fig. 8 (some nine orders of magnitude in the case of Petrel) is worthy of discussion. We point out first that set-up operations performed by Globus for each transfer result in a fixed startup cost of around one second. This fixed cost means that a transfer of size N bytes cannot achieve a rate faster than N bytes/sec, resulting in the clearly delineated ascending line of maximum observed performance as a function of transfer size in Figs. 8A, 8C and 8D: the upper envelope. Fortunately, this effect is not significant for the large transfers with which we are mostly concerned: for those, the performance of the MRDP’s network or storage system defines the upper bound. We attribute the somewhat less well delineated, lower envelope to the time that portal users are prepared to wait for transfers to complete.

For larger transfers, we know from other work (Liu et al., 2017; Rao et al., 2016) that excellent performance can be achieved when both the source and destination endpoints and the intervening network(s) are configured appropriately. Figure 8B, which extracts from Fig. 8A the transfers from RDA to NERSC, illustrates this effect. NERSC has a powerful network connection and array of DTNs, and RDA-to-NERSC transfers achieve rates of greater than 1 GB/s over the 10 Gbit/s RDA network connection. However, while the RDA and Petrel portals, in particular, manage data located on well-configured DTNs that are connected to high-speed networks without interference from intervening firewalls, a transfer may encounter quite different conditions en route to, or at, the corresponding source or destination. Low capacity networks, overloaded storage systems, firewalls, and lossy connections can all slash transfer performance (Liu et al., 2017). Failures in particular can play havoc with performance: the Globus transfer service works hard to recover from even long-duration network and storage system failures by retrying transfers over extended periods. A transfer is only recorded as complete when every byte has been delivered and verified correct. If errors have intervened, the reported transfer rate can be extremely low. These factors all contribute to a broad spectrum of observed transfer rates.

We comment on some specifics of the portals shown in the figures. The Sanger portal accepts user-uploaded datasets and returns processed datasets that are roughly an order of magnitude larger. Its transfer rates vary as it is used by individual users with varied connectivity. Petrel handles extremely large data and provides high-speed data access via its many DTNs and high-speed networks that connect it to users within Argonne and at other labs. The RDA portal primarily serves data of modest size (10s of GBs) and achieves mostly modest rates—although not in the case of NERSC, as noted above. Finally, Globus data publication is a unique portal in that its storage is distributed across 300 different Globus endpoints. Its transfer rate is therefore variable and usage is sporadic.

Related work

The MRDP design pattern that we have presented here codifies the experience of many groups who have deployed and applied research data portals. Our discussion of implementation techniques similarly reflects this experience. However, the design pattern could also be implemented with other technologies, as we now discuss.

The Globus transfer service typically employs the Globus implementation of the GridFTP transfer protocol (Allcock et al., 2005) for file transfers, which uses techniques such as multiple TCP channels (Hacker, Noble & Athey, 2004) to optimize transfer performance. Other protocols and tools that can be used for this purpose include Fast TCP (Jin et al., 2005), Scalable TCP (Kelly, 2003), UDT (Gu & Grossman, 2007), and bbcp (Hanushevsky, Trunov & Cottrell, 2001).

PhEDEx (Egeland, Wildishb & Huang, 2010) and the CERN File Transfer Service (Laure et al., 2006) are two examples of services that support management of file transfers. Both are designed for use in high energy physics experiments. The EUDAT (Lecarpentier et al., 2013) project’s B2STAGE service supports the transfer, via the GridFTP protocol, of research data between EUDAT storage resources and high-performance computing (HPC) workspaces. None of these systems provide APIs for integration with data portals and Science DMZs.

Apache Airavata (Pierce et al., 2015) provides general-purpose middleware for building science gateways, but does not address data movement or integration with Science DMZs. CyVerse (Goff et al., 2011) provides a scalable data store, built on iRODS (Rajasekar et al., 2010), and specialized APIs (Dooley et al., 2012) to access its storage services. It supports Cyberduck, amongst other tools, for transferring data using an array of protocols (e.g., HTTP, FTP, WebDAV) directly to Windows and Mac PCs. Thus, while it may separate control and data channels, it does not support third-party or high performance data transfer nor does it integrate with Science DMZs and DTNs.

Many services support scientific data publication, including Dataverse (Crosas, 2011), figshare, and Zenodo. Increasingly, these services are incorporating support for large amounts of data. For example, the developers of the Structural Biology Data Grid (Morin et al., 2013) are working to integrate Globus with Dataverse (Meyer et al., 2016). Similarly, publication repositories such as figshare now support data storage on institutional or cloud storage, thereby achieving some separation of control and data channels. While these implementations follow some aspects of the MRDP design pattern, they do not integrate with Science DMZs and their proprietary implementations cannot be easily generalized and adopted by others.

The MRDP design pattern could also be implemented using commercial cloud services, such as those provided by Amazon, Google, and Microsoft. Indeed, many public scientific datasets are now hosted on cloud platforms and researchers are increasingly leveraging cloud services for managing and analyzing data (Babuji et al., 2016). Cloud services provide scalable data management, rich identity management, and secure authentication and authorization, all accessible via APIs—a complete set of capabilities needed to implement the MRDP design pattern. However, each cloud provider is a walled garden, not easily integrated with other systems, and cloud provider business models require that MRDP administrators develop methods to recoup service costs related to data storage and egress. In contrast, our MRDP reference implementation allows data to be stored and retrieved from many locations.

Summary

We have described the state of the practice for delivering scientific data through what we call the modern research data portal (MRDP) design pattern. The MRDP pattern addresses the shortcomings of the monolithic legacy research data portal by improving transfer performance, security, reliability, and ease of implementation while also reducing operational complexity.

We have shown how high performance networks and cloud service APIs can be used to create performant and particularly simple implementations of this design pattern. In a typical deployment, as shown in Fig. 3, the control logic of Listing 1 runs on a protected computer behind the institutional firewall, to protect that sensitive logic against attack, while the storage system(s) on which data reside sit inside the Science DMZ, with Globus endpoint(s) deployed on DTNs for high-speed access. The control logic makes REST API calls to the Globus cloud service to create shared endpoints, transfer files, manage permissions, and so forth.

The sample code we have presented shows how developers can automate data management tasks by using modern cloud-based data services to create powerful research data portals (accessible via Web, mobile, custom applications, and command line) that leverage Science DMZ paths for data distribution, staging, replication, and other purposes. In our example we leverage Globus to outsource all identity management and authentication functions. The MRDP needs simply to provide service-specific authorization, which can be performed on the basis of identity or group membership. And because all interactions are compliant with OAuth 2 and OpenID Connect standards, any application that speaks these protocols can use the MRDP service like they would any other; the MRDP service can seamlessly leverage other services; and other services can leverage the MRDP service. The use of best-practice and standards-compliant implementations for data movement, automation, authentication, and authorization is a powerful combination.

The benefits of the MRDP approach lie not only in the separation of concerns between control logic and data movement. In addition, the data portal developer and admin both benefit from the ability to hand off the management of file access and transfers to the Globus service. In the last three years we have observed steady adoption of the MRDP design pattern. We described five illustrative implementations that variously serve research data, support analysis of uploaded data, provide flexible data sharing, enable data publication, and facilitate data archival. Collectively, these deployments have performed more than 80,000 transfers totaling almost two petabytes over the past three years.

We thank the Globus team for the development of the technologies described here, and participants in “Building the Modern Research Data Portal” workshops for their feedback.

Additional Information and Declarations

Competing Interests

Author Contributions

Data Availability

Ian Foster is an Advisor and Academic Editor for PeerJ Computer Science.

Kyle Chard and Ian Foster conceived and designed the experiments, analyzed the data, contributed reagents/materials/analysis tools, wrote the paper, prepared figures and/or tables, performed the computation work, reviewed drafts of the paper.

Eli Dart conceived and designed the experiments, analyzed the data, wrote the paper, prepared figures and/or tables, reviewed drafts of the paper.

David Shifflett and Jason Williams performed the experiments, performed the computation work, reviewed drafts of the paper.

Steven Tuecke conceived and designed the experiments, analyzed the data, wrote the paper, prepared figures and/or tables, performed the computation work, reviewed drafts of the paper.

The following information was supplied regarding data availability:

The companion web site, http://docs.globus.org/mrdp, provides references to GitHub for associated code.

Github: https://github.com/globus/globus-sample-data-portal for the code.

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
