# Peer review of "The Modern Research Data Portal: a design pattern for networked, data-intensive science"

_PeerJ Computer Science, doi:10.7717/peerj-cs.144_

## Round 0.1 · original submission · Minor Revisions

The reviewers all like your paper and agree that it should be published. The reviewers also have a few questions and/or suggestions, that I'd like you to address in your minor revision. In your response, please include how you've addressed the reviewers' comments, after which the paper will be accepted and published.

Reviewer 1 ·

Basic reporting

I note a few minor problems that, if fixed, would help with readability.

• The term “science DMZ” is introduced at line 49 but not explained until line 145. It would be helpful to insert a footnote to the first occurrence to replace the latter explanation.
• Similarly, Fig. 1 references perfSONAR but perfSONAR is not explained until line 158, and there only refers to Fig. 2.
• I believe the code snippet following line 381 is missing a closing parenthesis
• The text gets a bit repetitive in the explanations of authorization
• In Table 1, the units for Rate are given as mbps. It should at least be Mbps, or perhaps MBps – it is not clear.

Experimental design

The paper is not reporting on an experiment, but rather on the design and deployment of a research data portal.

Validity of the findings

This is a very valuable description of a software infrastructure that supports data discovery, transfer, and dissemination.

Additional comments

There is little to complain about in this manuscript from Chard et al. on the Modern Research Data Portal. The paper is well-written and timely, representing further fine work by the well-regarded team behind Globus. It is also commendable that the authors have made the code for their reference implementation freely available.

I find it hard to understand the transfer rate data shown in Fig. 8. These distributions show transfer rates that differ by six orders of magnitude! And as slow as ~100 bytes/sec, which is more like old acoustic-coupled modem transfer rates. I understand the upper envelope, in that you cannot exceed a transfer rate whose numerator is larger than the total amount of data transferred. But I do not understand the lower envelope unless there is some cut-off or time-out being applied such that transfers that take too long are aborted (and not counted). Also, it might make more sense to label the x-axis in powers of ten incremented by 3, e.g., 10^0, 10^3, 10^6, so that they map more obviously onto KB, MB, GB, etc.. Lastly, for Fig. 8a it is remarked that there is only one incoming transfer, but it is not at all visible in the plot.

Given that several of the authors are active in the National Data Services Consortium, I was disappointed that NDS is not mentioned, e.g., as a deployment platform for the reference implementation. One could imagine community engagement through customization of the reference implementation and sharing of those customizations through the NDS Labs Workbench.

Reviewer 2 ·

Basic reporting

The article is well written in clear, professional language with a good introduction and relevant set of references.

One comment on Figure 1 is the lack of explanation of the perfSONAR network monitoring devices. These are explained in the MRDP Figure 2 but are either unnecessary for the LRDP diagram in Figure 1 - they would surely not have been present in most legacy implementations – or should be explained.

Experimental design

The research issues are well explained and very relevant to the management and movement of the increasingly large amounts of experimental data being generated in large-scale facilities.

The code examples are well explained and relevant and the physical exemplars are interesting and well-chosen.

Validity of the findings

The evaluation of MRDP adoption is brief but very interesting.

The only example I would question is the inclusion of the Cornell Advanced Computing (CAC) archive. This was not described as an example of an MRDP design pattern and seems to be a different case. Does the Cornell high-speed internal network separate data transfer from general network traffic? If so, that would be very interesting but this was not made clear.

Additional comments

The paper is clearly written with interesting and relevant examples. There is now a real need for something like the MRDP design pattern and this paper shows how the necessary elements for authentication and authorization, data file management and fast data movement can be achieved with Science DMZs and a service like Globus Online.

I have made a couple of comments for the authors to consider clarifying but these should not prevent publication of this useful and important paper.

·

Basic reporting

Chard et al present a distributed systems software engineering pattern for building data intensive science portals.

This work presented is not necessarily new. However, this is the culmination of much work in this space by the Globus team, and other groups developing software infrastructure for global science. This publication to formalise this pattern, present real world examples, and demonstrate example code is very worthwhile and and very much worthy of publication as a complete whole.

The publication is well written and provides both a code walkthrough, example data portal and Jupyter notebook example.

Experimental design

The authors present a model that formalises the intersection of the science DMZ, high performance file transfer services, HTTP portals. The illustrate the success of this model by demonstrating successful implementations and providing statistics about their uptake, and allow authors to test and reuse by providing code and examples. I consider the experimental design and demonstration extensive for this type of publication.

Validity of the findings

The case made by the authors is valid and demonstrated.

I would suggest that using the terms “legacy research data portal” and “modern research data portal” is loaded and the authors may want to reconsider this naming.

---

## Round 0.2 · accepted · Accept

Thank you again for your responses and clarifications in the rebuttal letter, and submitting a revised version!

... and since it's that week of the year: Happy Holidays!